# Ecuadorian Provinces with High Morbidity and Mortality Rates Due to Asthma among the Working-Age Population: An Ecological Study to Promote Respiratory Health

**DOI:** 10.3390/ijerph21070909

**Published:** 2024-07-11

**Authors:** Antonio Ramón Gómez-García, Andrea Liseth Cevallos Paz, Diemen Delgado-Garcia, Danilo Martínez Jimbo

**Affiliations:** 1Facultad de Postgrados, Universidad Espíritu Santo, Samborondón 092301, Ecuador; 2Ecuadorian Observatory of Occupational Safety and Health, Samborondón 092301, Ecuador; 3Department of Occupational Safety and Health, Occupational Medicine-PraxMED, Quito 170135, Ecuador; andicp91@hotmail.com; 4Universidad de Aconcagua, San Felipe 2170000, Chile; diemendelgadogarcia@gmail.com; 5International Pneumoconiosis Observatory, Santiago de Chile 7500494, Chile; 6Universidad Internacional SEK, Quito 170134, Ecuador; danilo.martinez@uisek.edu.ec; 7Ecuadorian Society of Occupational Medicine, Quito 170519, Ecuador

**Keywords:** asthma, working-age population, geographic differences, respiratory diseases, public health

## Abstract

Asthma is a significant public health concern. This study identified the provinces with the highest morbidity and mortality rates due to asthma among the working-age population (15–69 years) in the Republic of Ecuador. The secondary objective was to explain the possible differences attributable to occupational exposure. This nationwide ecological study was conducted in 24 provinces between 2016 and 2019. Government databases were used as sources of information. Age-standardized rates were calculated for codes J45 and J46. The hospitalization morbidity rate for asthma decreased from 6.51 to 5.76 cases per 100,000 working-age population, and the mortality rate has consistently been low and stable from 0.14 to 0.15 deaths per 100,000 working-age population. Geographic differences between the provinces were evident. The risk of hospitalization and death due to asthma was higher in the Pacific coast (Manabí with 7.26 and 0.38, Esmeraldas with 6.24 and 0.43, Los Ríos with 4.16 and 0.40, El Oro with 7.98 and 0.21, Guayas with 4.42 and 0.17 and the Andean region (Azuay with 6.33 and 0.45, Cotopaxi (5.84 and 0.48)). The high rates observed in provinces with greater agricultural and industrial development could be national heterogeneity’s main determinants and act as occupational risk factors. The contribution of occupational hazards in each province should be examined in depth through ad hoc studies. The findings presented here provide valuable information that should prompt further detailed studies, which will assist in designing public policies aimed at promoting and safeguarding the respiratory health of the population, particularly that of workers. We believe that this study will inspire the creation of regional networks for the research and surveillance of occupational health.

## 1. Introduction

Asthma is a non-communicable disease characterized by the narrowing of the airways, which can affect individuals of all age groups [1,2]. The global prevalence of asthma has increased significantly in recent decades, while mortality rates have decreased. In 2019, it was estimated that approximately 262 million individuals were affected by asthma, with 455,000 fatalities attributed to the condition [3]. Consequently, asthma is regarded as a significant global public health concern due to its adverse impact on quality of life and premature mortality [4].

Although the age at which asthma develops can vary, it is commonly observed in young adults, who are part of the labor-productive population. When asthma is associated with working conditions, it can be considered an occupational health concern [5,6]. Occupational asthma may be precipitated by exposure to many agents in the workplace, including chemical and biological substances that act as sensitizing and irritating substances [7]. It has been demonstrated that certain economic sectors situated in specific geographical areas may be more prone to developing asthma among workers. For example, this problem has been identified in workers in both the industrial and the agricultural sectors [8]. The study of asthma disparities between different regions of the same country is therefore of particular interest [9,10], as it could provide valuable information for the targeted investment of health resources in the prevention and treatment of respiratory diseases among workers [11].

In the Republic of Ecuador (hereafter referred to as “Ecuador”), there is a lack of studies on asthma morbidity and mortality rates in the working-age population, especially when disaggregated by territorial level. Another knowledge gap is the under-reporting of occupational diseases among workers covered by the social security system [12]. Between 2016 and 2019, the only years for which records were available, there were 64 reports of occupational asthma in Ecuador. The provinces with the highest concentration of occupational asthma cases were Pichincha (35.9%) and Guayas (28.1%) (see Appendix A). There is probably a hidden reality in the rest of the country’s provinces. The failure to recognize and address this hidden reality could result in the development of ineffective policies, which would hinder the ability of the government to protect the respiratory health of workers and could lead to the worsening of the current situation in the provinces.

This study aimed to identify the Ecuadorian provinces with the highest rates of asthma morbidity and mortality in the working-age population and examine possible geographical differences in relation to environmental factors and occupational risks.

## 2. Materials and Methods

### 2.1. Context and Population

Ecuador is located in northern South America and encompasses an area of approximately 256,370 km^2^. This country is renowned for its rich cultural diversity and significant climatic variation due to its geographical location. In 2019, the country had a population of approximately 17.3 million, with the urban areas of the provinces of Pichincha (with the capital, Quito) and Guayas comprising a significant proportion of the population (43.4%). The service sector is most prominent in most provinces. However, it is noteworthy that mining and oil extraction are significant in the Amazon region, while agricultural-related activities are prominent in the Andean region and the Pacific coast.

The working-age population in this study was between 15 and 69 years. This age group substantially contributes to the economy. In this country, the working-age population totals 11.9 million persons, representing 68.8% of the entire population (2019), with 51% women and 49% men.

### 2.2. Design, Data Sources, and Statistical Analysis

An observational ecological study was conducted across 24 provinces, extending to the Galapagos Islands, using the administrative records of hospital care and medical certificates of death from 2016 to 2019. Information was obtained from the Statistical Register of Hospital Beds and Emergencies [13], which records the number of individuals treated within the Integrated Public Health Network, and the General Defunction Register [14], which documents the causes of death based on the International Classification of Diseases (ICD-10). Both sources of information are compiled by the National Institute of Statistics and Census of Ecuador (known by its Spanish acronym INEC) and undergo statistical quality control before being published on the web (Open Access Database, https://www.ecuadorencifras.gob.ec, accessed on 21 January 2024). The INEC is responsible for regulating, planning, directing, coordinating, and supervising the country’s official statistics.

Codes J45 [asthma: predominantly allergic asthma (J45.0), nonallergic asthma (J45.1), mixed asthma (J45.8), and unspecified (J45.9)] and J46 (status asthmaticus) were selected. These codes have been used in previous studies to quantify the number of deaths and work-related asthma diseases resulting from exposure to hazardous chemicals and air, water, and soil pollution in occupational environments [12,15,16].

For the data extraction, the province of residence was considered, assuming it was also the place of work. Foreign residents (nine diagnosed cases and no deaths) were excluded from the analysis. From 2016 to 2019, 2786 patients were diagnosed, and 64 individuals died.

The estimated annual asthma morbidity and mortality rates are expressed per 100,000 person-years for selected codes. After calculating age-specific rates, to compare provinces and control for the effect of population composition, age-standardized rates (ASRs) and their corresponding confidence intervals (95% CI) were calculated using the direct method, taking into account the Ecuadorian population standard for defined ages from 15 to 69 years [17]. The denominator used for the ASR was obtained from the population projections of the National Institute of Statistics and Censuses (see Appendix A).

Due to the limited duration of the four-year period in question, linear regression analysis was used to show statistically significant changes in the slopes of the ASR (increasing or decreasing). Finally, a quadrant chart was used to illustrate each province’s average asthma mortality and morbidity rates. The results were subjected to an analysis to identify potential explanations for the environmental factors and occupational risks.

## 3. Results

The hospitalization morbidity rate for asthma in Ecuador decreased from 6.51 to 5.76 cases per 100,000 working-age population between 2016 and 2019. However, no significant reduction was observed (slope = −0.28, *p* = 0.122). The asthma mortality rate has been low and consistently stable (slope = 0.01, *p* = 0.599), with values ranging between 0.14 and 0.15 deaths per 100,000 working-age population.

### 3.1. Morbidity by Provinces

Table 1 presents the prevalence and annual rates of hospital morbidity for asthma among the working-age population across the 24 provinces of the national territory. The provinces with the highest average prevalence were Pichincha (19.7%), Guayas (18.2%), and Manabí (10.4%).

Nonetheless, when calculating the ASR, eight provinces reported higher morbidity rates for asthma than the national average (7.82 cases per 100,000 inhabitants). The most significant provinces in the Amazon region are Zamora Chinchipe, Morona Santiago, Sucumbíos, and Loja in the Andean region. According to the statistical analysis, El Oro province decreased from 9.68 (95% CI, 7.03–12.99) to 55.76 (95% CI, 3.83–8.33) cases per 100,000 working-age population (slope = −1.34, *p* = 0.018), while Chimborazo province increased from 5.72 (95% CI, 3.38–9.04) to 9.35 (95% CI, 6.35–13.28) cases per 100,000 working-age population (slope = 1.17, *p* = 0.008).

### 3.2. Mortality by Provinces

Regarding asthma mortality, some provinces did not report any cases during the study period, whereas others only reported cases in some years. Table 2 shows only those provinces that reported asthma-related deaths over consecutive years.

Guayas (29.8%) and Manabí (24.3%) had the highest average number of asthma-related deaths due to asthma. The province of Los Ríos, located in the Pacific region, was distinguished by its high mortality rate from asthma, which remained above the national average throughout the period (0.36 deaths per 100,000 working-age population). The province of Esmeraldas was notable for its significant increase between 2017 and 2018, although no cases were reported in the following year.

### 3.3. The Identification of the Provinces with the Highest Morbidity and Mortality Rates

Individual rates alone are insufficient for identifying the most problematic provinces. In accordance with our methodology, we employed a quadrant chart to categorize the provinces based on their combined morbidity and mortality ASRs. This classification was applied only to provinces that had reported cases for more than two consecutive years, as depicted in Figure 1.

## 4. Discussion

The current investigation results reveal disparities that contrast with global patterns in the prevalence of asthma morbidity and mortality [4] and challenge those documented in other settings [18]. In our study, we observed a decrease in the morbidity rate and an increase in the mortality rate in the age range of 15–69 years. Differences between the provinces of Ecuador were evident from 2016 to 2019.

As previously stated, the observed differences may be attributed to many specific factors unique to each region [10], even within the distinctive characteristics of each territory within our country. Given the intricacy of establishing more precise causal relationships and the paucity of supplementary data, we present plausible explanations for the observed discrepancies between the provinces identified in this study. To address these issues, two blocks were considered: climate and environmental factors, as well as the specific occupational risks present in the working conditions of each province.

### 4.1. Environmental and Geoclimatic Factors

Several studies have associated the development of asthma with mold and fungal exposure in the home environment [19,20]. Regarding the above quadrant (Figure 1), the provinces of Esmeraldas, Manabí, and El Oro (including Guayas) are located in the coastal region of the country and are characterized by a humid tropical climate with an average annual temperature ranging between 24 and 31 °C, as well as rainy seasons from December to April–May. It can be assumed that these climatic particularities increased the likelihood of hospital care during certain seasons. In contrast, a significant association between low temperature and asthma has been demonstrated [21]. In addition, the provinces of Azuay are located in the Andes Mountains, and because of their altitude (more than 2500 feet above the mean sea level), temperatures can decrease to 6 °C.

Furthermore, it is important to note that various studies have identified a higher prevalence of asthma in the urban areas of regions with a higher population density and consequent exposure to environmental pollution [6,22]. Guayas and Pichincha were the two most populous and industrialized provinces in the country, with the city of Guayaquil (Guayas) representing a particularly notable concentration of population and industry.

### 4.2. Attributable Occupational Factors

Asthma is a complex disease that can be diagnosed and influenced by several factors. In particular, the occupational risk of each economic activity plays a key role. The Azuay and Guayas provinces have the largest number of companies in the industrial sector. It has been widely demonstrated that workers exposed to chemical agents have a higher prevalence and risk of developing occupational asthma induced by both sensitizers and irritants [6,7]. Additionally, agro-industrial activities predominated in El Oro and Los Ríos (in addition to Guayas). Similarly, exposure to insecticides, fungicides, and herbicides can cause asthma in the working-age population [9].

In our study, Orellana (increase in morbidity from 2.46 in 2016 to 6.83 in 2019) and Santa Elena (increase in morbidity from 3.49 in 2016 to 5.25 in 2019) were not among the provinces with the highest morbi-mortality from asthma; they stand out as the only provinces with an increase in morbidity rates. Notably, there are two oil refineries in these provinces. It is presumed that exposure (environmental and occupational) to the chemicals used during refining processes could influence this fact [23]. The province of Esmeraldas, which has the second-highest mortality rate and the largest refinery in the country, would leave less uncertainty.

Finally, the province of Manabí (Manta city) has experienced remarkable urban developments in recent years. It is plausible that the high morbidity from asthma may also be due to the inhalation of wood, silica, asbestos, and solvents, as well as exposure to welding smoke among construction workers [24].

In addition, there are several reasons that could explain the differences in asthma morbidity and mortality between the different provinces of the country. First, from a technical point of view, companies may not provide their workers with adequate respiratory protection, which would significantly reduce the risk of developing asthma due to exposure to occupational hazards. Second, from a health surveillance perspective, companies may be overlooking the need to adequately monitor the respiratory health of their workers for early detection and intervention. Third, the lack of visits by labor inspectors to enterprises in the most affected provinces could be another reason. 

### 4.3. Opportunities for Respiratory Health Promotion

It should be noted that this study was subject to certain limitations that must be considered. Firstly, the utilization of secondary sources of information precluded the conduct of more rigorous analyses, which would have enabled the identification of the most affected economic sectors and occupations. This information would have been helpful in identifying the most vulnerable workers [8]. Although the quality of administrative records has improved in recent years, there may be an important infradiagnosis that exists in this context [18]. Furthermore, it is essential to note that we did not find a correlation between the standardized morbidity rates and asthma mortality in the identified provinces.

Our results are similar to those of previously carried out studies, corroborating the decreasing trend in hospital admissions and deaths from asthma between 2000 and 2018 in large regions of the country [18]. Nevertheless, it is challenging to make comparisons with other studies, given that our study has conducted a more precise analysis at the provincial level and for a specific population group.

This finding suggests a need to counteract future research hypotheses that help explain it. These limitations do not alter the findings regarding geographical heterogeneity or the behavior of the observed trends. This study provides insights into the importance of this respiratory disease and provides the first approach to the current situation in the country for the working-age population [25,26]. Additionally, our results provide a breakdown analysis by provincial unit and updated knowledge from previous research [27].

## 5. Conclusions

In conclusion, this study makes a noteworthy contribution by highlighting the insufficient understanding of asthma among working-age groups in Ecuador at a regional level. Environmental factors, including climatic conditions and pollution, and occupational exposure, particularly in the industrial and agro-industrial sectors, play important roles in the prevalence of asthma. It is recommended that geographic identification in provinces with high morbidity and mortality rates be prioritized in implementing prevention and respiratory health promotion programs [28].

There is an urgent need to slow the normal decline with aging or in response to common or occupational exposures as an essential part of any primary prevention initiative. Nevertheless, further research is required to gain a more comprehensive understanding of this issue and to provide more precise information, which will in turn facilitate the design of an appropriate public health and occupational health action plan to tackle asthma morbidity and mortality.

## Figures and Tables

**Figure 1 ijerph-21-00909-f001:**
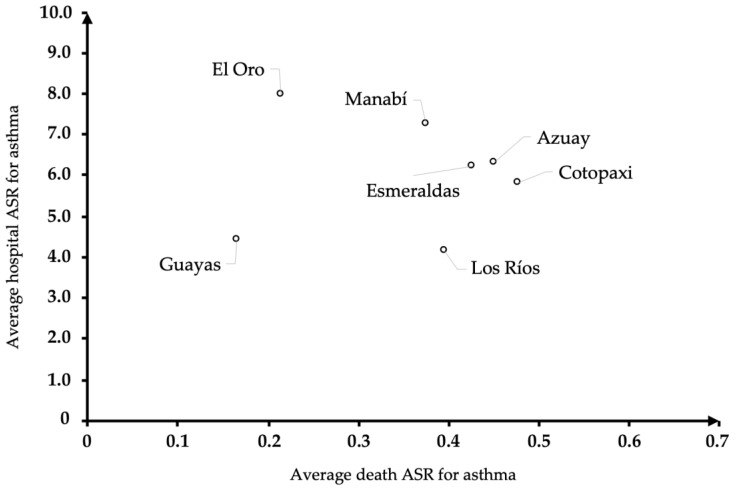
The identification of the provinces with the highest morbidity and mortality rates due to asthma in Ecuador during the period from 2016 to 2019. Source: author’s own elaboration from the General Register of Deaths and Hospital Highs [13,14].

**Table 1 ijerph-21-00909-t001:** Prevalence and asthma morbidity rates in working-age population by province in Ecuador from 2016 to 2019.

	2016 (N = 702)				2017 (N = 735)				2018 (N = 691)				2019 (N = 658)				Slope	*p*
Province	%	Crude	ASR (95% CI)		%	Crude	ASR (95% CI)		%	Crude	ASR (95% CI)		%	Crude	ASR (95% CI)			
Azuay	4.7	6.09	6.18	(4.25–8.68)	4.6	6.13	6.12	(4.24–8.55)	5.4	6.52	6.51	(4.58–8.97)	5.8	6.55	6.50	(4.60–8.93)	0.13	0.156
Bolívar	1.3	7.56	7.38	(3.36–14.04)	1.1	6.64	7.09	(3.04–14.00)	1.3	7.37	7.70	(3.5–14.7)	1.5	8.09	8.43	(4.03–15.52)	0.38	0.158
Cañar	1.1	4.93	5.17	(2.22–10.19)	1.0	4.22	4.06	(1.62–8.39)	1.0	4.12	4.31	(1.72–8.89)	2.0	7.48	7.83	(4.15–13.41)	0.82	0.384
Carchi	-				0.4	2.51	2.37	(0.48–6.94)	1.6	9.09	8.85	(4.41–15.85)	0.5	2.45	2.38	(0.48–6.95)	1.36	0.538
Cotopaxi	3.0	7.38	7.54	(4.66–11.52)	1.6	4.14	4.10	(2.11–7.16)	2.3	5.41	5.60	(3.20–9.09)	2.7	5.98	6.14	(3.63–9.71)	−0.27	0.755
Chimborazo	2.6	5.65	5.72	(3.38–9.04)	3.0	6.81	6.90	(4.32–10.45)	3.6	7.62	7.74	(5.01–11.43)	4.7	9.31	9.35	(6.35–13.28)	1.17	0.008
El Oro	6.3	9.69	9.68	(7.03–12.99)	5.7	9.08	9.08	(6.54–12.27)	5.1	7.43	7.40	(5.10–10.30)	4.3	5.84	5.76	(3.83–8.33)	−1.34	0.018
Esmeraldas	3.0	5.74	5.92	(3.66–9.05)	3.0	5.88	6.23	(3.90–9.43)	3.8	6.81	7.00	(4.60–10.30)	3.3	5.64	5.80	(3.63–8.79)	0.04	0.895
Galápagos	0.4	14.72	15. 9	(3.15–46.87)	0.3	9.58	10.05	(1.07–36.44)	0.1	4.68	4.38	(0.06–24.38)	0.2	4.57	4.42	(0.06–24.58)	−4.04	0.058
Guayas	16.5	4.17	4.17	(3.44–5.00)	18.9	4.91	4.88	(4.10–5.76)	17.5	4.19	4.20	(3.48–5.02)	19.9	4.46	4.45	(3.72–5.28)	0.02	0.937
Imbabura	2.1	5.18	5.19	(2.90–8.56)	2.6	6.43	6.50	(3.91–10.15)	1.4	3.32	3.30	(1.60–6.15)	2.4	5.20	5.19	(2.97–8.43)	−0.32	0.686
Loja	6.7	14.90	14.75	(10.83–19.62)	10.2	23.39	23.72	(18.65–29.74)	5.6	11.97	12.10	(8.63–16.61)	3.3	6.65	6.75	(4.23–10.22)	−3.56	0.351
Los Ríos	2. 3	2.87	2.88	(1.65–4.68)	3.9	5.10	5.12	(3.42–7.35)	4.5	5.36	5.38	(3.66–7.64)	2.9	3.22	3.24	(1.95–5.06)	0.13	0.865
Manabí	14.0	10.03	9.97	(8.10–12.16)	8.3	6.15	6.10	(4.67–7.84)	10.1	6.95	6.96	(5.42–8.79)	9.3	5.97	6.00	(4.59–7.71)	−1.11	0.233
Morona Santiago	3.3	22.55	23.57	(14.79–35.56)	3.8	26.51	28.33	(18.66–41.16)	2.5	15.55	15.96	(9.18–25.70)	1.2	7.07	7.72	(3.25–15.34)	−5.99	0.143
Napo	0.6	5.47	6.07	(1.59–15.62)	0.4	3.99	3.84	(0.73–11.32)	0.7	6.45	7.23	(2.31–16.90)	0.5	3.76	4.02	(0.80–11.78)	−0.28	0.783
Orellana	0.3	2.24	2.46	(0.26–8.95)	0.5	4.39	4.19	(1.13–10.73)	0.9	6.46	7.19	(2.58–15.75)	0.9	6.33	6.83	(2.45–14.95)	1.61	0.074
Pastaza	0.6	6.37	6.61	(1.70–17.10)	0.3	3.07	3.19	(0.31–11.67)	0.1	1.48	1.83	(0.02–10.17)	0.8	7.16	7.74	(2.46–18.12)	0.20	0.906
Pichincha	20.8	7.14	7.07	(5.97–8.32)	18.1	6.36	6.29	(5.26–7.45)	18.7	6.04	5.95	(4.97–7.07)	21.3	6.42	6.31	(5.31–7.45)	−0.26	0.285
Santa Elena	1.1	3.47	3.49	(1.50–6.88)	2.2	6.76	6.67	(3.81–10.83)	2.0	5.76	5.60	(3.06–9.39)	2.0	5.21	5.25	(2.79–8.98)	0.42	0.589
Santo Domingo T.	1.0	2.55	2.54	(1.01–5.24)	1.1	2.84	2.78	(1.20–5.48)	2.0	4.85	5.00	(2.73–8.39)	2.1	4.73	4.85	(2.65–8.15)	0.91	0.100
Sucumbíos	2.3	12.08	12.41	(7.06–20.19)	2.0	10.98	11.85	(6.60–19.59)	2.0	9.95	10.47	(5.69–17.61)	1.1	4.83	5.33	(2.12–10.99)	−2.26	0.095
Tungurahua	4.4	8.25	8.16	(5.55–11.59)	4.2	8.12	7.98	(5.42–11.33)	4.6	8.25	8.14	(5.57–11.49)	5.9	9.90	9.71	(6.90–13.28)	0.48	0.236
Zamora Chinchipe	1.7	18.41	19.57	(10.02–34.31)	2.7	29.78	31.91	(19.40–49.39)	3.0	30.36	32.25	(19.86–49.43)	1.5	14.04	13.78	(6.51–25.48)	−1.70	0.761
Ecuador	25.2	6.51			26.38	6.68			24.8	6.16			23.62	5.76			−0.28	0.122

*p* <  0.001 (Bonferroni test). ASR, age-standardized rate. Source: author’s elaboration from the Hospital Highs [13].

**Table 2 ijerph-21-00909-t002:** Prevalence and asthma mortality rates in working-age population by province in Ecuador from 2016 to 2019.

	2016 (N = 15)				2017 (N = 10)				2018 (N = 22)				2019 (N = 17)				Slope	*p*
Province	%	Crude	ASR (95% CI)		%	Crude	ASR (95% CI)		%	Crude	ASR (95% CI)		%	Crude	ASR (95% CI)			
Azuay	-				-				13.6	0.53	0.57	(0.11–1.65)	11.8	0.34	0.33	(0.04–1.19)	0.15	0.660
Cotopaxi	6.7	0.35	0.36	(0.005–2.00)	-				9.1	0.68	0.72	(0.08–2.60)	5.9	0.33	0.35	(0.005–1.97)	0.07	0.697
El Oro	6.7	0.22	0.22	(0.003–1.20)	-				4.5	0.21	0.21	(0.003–1.18)	5.9	0.21	0.21	(0.003–1.16)	0.02	0.782
Esmeraldas	-				10.0	0.27	0.29	(0.004–1.60)	9.1	0.52	0.56	(0.06–2.01)	-				0.28	0.013
Guayas	40.0	0.22	0.21	(0.08–0.46)	20.0	0.07	0.07	(0.01–0.25)	18.2	0.14	0.14	(0.04–0.35)	41.2	0.24	0.24	(0.10–0.50)	0.02	0.728
Los Ríos	13.3	0.36	0.36	(0.04–1.31)	30.0	0.53	0.52	(0.11–1.53)	9.1	0.35	0.35	(0.04–1.27)	11.8	0.34	0.35	(0.04–1.25)	−0.02	0.691
Manabí	26.7	0.41	0.41	(0.11–1.06)	30.0	0.30	0.30	(0.06–0.88)	22.7	0.50	0.50	(0.16–1.16)	17.6	0.29	0.29	(0.06–0.85)	−0.02	0.792
Ecuador	23.44	0.14			15.63	0.09			34.38	0.20			26.56	0.15			0.01	0.599

*p*-value = 0.403 (Bonferroni test). ASR, age-standardized rate. No deaths occurred in the following provinces: Bolívar, Cañar, Carchi, Chimborazo, Galápagos, Morona Santiago, Napo, Orellana, Pastaza, Santo Domingo, Sucumbíos, Tungurahua, and Zamora Chinchipe. Provinces with death records in a single year. Imbabura reported a total of 10.0% of deaths in 2017, with a crude death rate of 0.34 and an ASR of 0.35 (95% CI, 0.005–1.94) per 100,000 working-age population. Loja reported a total of 9.1% of deaths in 2018, with a crude death rate of 0.61 and an ASR of 0.61 (95% CI, 0.07–2.19) per 100,000 working-age population. Santa Elena reported a total of 5.9% of deaths in 2019, with a crude death rate of 0.40 and an ASR of 0.42 (95% CI, 0.01–2.33) per 100,000 working-age population. In 2016 and 2018, there were 6.7% of deaths, a crude death rate of 0.05, and an ASR of 0.05 (95% CI, 0.001–0.26) per 100,000 working-age population and 4.5% of deaths, a crude death rate of 0.05, and an ASR of 0. 05 (95% CI, 0.001–0.25) per 100,000 working-age population. Source: author’s elaboration from the General Register of Deaths [14].

## Data Availability

All studies in our study are included in the Appendix A.

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
