# Peer review of "Ecuadorian Provinces with High Morbidity and Mortality Rates Due to Asthma among the Working-Age Population: An Ecological Study to Promote Respiratory Health"

_ijerph, 2024, doi:10.3390/ijerph21070909_

Round 1

Reviewer 1 Report

Comments and Suggestions for Authors

The manuscript entitled "Ecuadorian Provinces with High Morbidity and Mortality Rates due to Asthma among Working-Age Population: An Ecological and Exploratory Study to Promoting Respiratory Health," has presented the population based study relying on the demographic data from the National Databases and has meticulously described the prevalence of Asthma leading to high mortality and morbidity rates among the different age groups and also tried to shed light on the underlying reasons. To qualitatively enhance the manuscript for publication some major revisions are necessary, which are as follows:

1.     Though the abstract provides a succinct overview of the manuscript but the precise statistics and significant findings should be included rather than generalized statements. Moreover highlight the implications of the findings for formulation of explicit public policy.

2.     In introduction clearly state the gap this study addresses and its significance in the field of public health.

3.     In the method section, explain why an ecological and exploratory study design was chosen and also provide the details of the databases. Elaborate the statistical methods used for the specific data analysis.

4.     Provide a more detailed discussion of the differences in asthma morbidity and mortality across provinces. Discuss possible reasons for the observed trends in specific regions.

5.     Perform a comparative analysis with the other epidemiological studies performed on Asthma mortality and morbidity in Ecuador region.

6.     Authors should try to provide a forecast and a proper action plans to deal with the morbidity and mortality that can be a torchbearer for the formulations of Government policies.

Reviewer 2 Report

Comments and Suggestions for Authors

Asthma is a major noncommunicable disease, affecting both children and adults. Although it can be a serious condition, asthma can be managed with the right treatment. However, asthma is often under-diagnosed and under-treated, particularly in low- and middle-income countries.

In this study, Antonio Ramón Gómez-García et al. conducted a nationwide ecological and exploratory study of asthma across 24 Ecuadorian provinces between 2016 and 2019, using government databases as sources of information. This study makes a noteworthy contribution by highlighting the insufficient understanding of asthma among working-age groups in Ecuador at a regional level.

In general, this study provides valuable information that could be used to create public policies aimed at promoting and safeguarding the respiratory health of the population, particularly workers. However, I have one concern regarding the methodology. The statistical analysis presented in Tables 1 and 2 did not include multiple testing corrections. Given that this study aims to provide information for creating public policies, more rigorous statistical methods should be applied to ensure the reliability of the findings.

Comments on the Quality of English Language

Extensive editing of English language required

Round 2

Reviewer 1 Report

Comments and Suggestions for Authors

The authors have answered all the queries I raised and improved the manuscript. The revised manuscript is recommended for publication.

Reviewer 2 Report

Comments and Suggestions for Authors

I have no further comments.

Comments on the Quality of English Language

Minor editing of English language required